# Background of Minimally Invasive Glaucoma Surgery (MIGS)-Adapted Patients for Cataract Surgery in Glaucoma

**DOI:** 10.3390/jcm13185378

**Published:** 2024-09-11

**Authors:** Yasunari Hayakawa, Takayuki Inada

**Affiliations:** 1Urawa Central Eye Institution, Saitama 336-0042, Japan; 2Kawagoe Central Eye Institution, Kawagoe 350-1122, Japan; 3Koshigaya Central Eye Institution, Koshigaya 343-0041, Japan; 4Kumagaya Central Eye Institution, Kumagaya 360-0833, Japan; 5Yorii Central Eye Institution, Yorii 369-1202, Japan; 6Onohara Eye Institution, Chichibu 368-0005, Japan; 7Kisarazu Central Eye Institution, Kisarazu 292-0823, Japan

**Keywords:** glaucoma, minimally invasive glaucoma surgery, cataract, intraocular pressure

## Abstract

**Purpose**: To investigate whether it is better to perform minimally invasive glaucoma surgery (MIGS) when performing cataract surgery on glaucoma patients. **Methods**: The study enrolled a total of 159 eyes of subjects with mild-to-moderate glaucoma, including primary open angle glaucoma (POAG), normal tension glaucoma (NTG), and combined mechanism glaucoma (CMG) with visually significant cataract, who were treated with one or more ophthalmic antiglaucoma agents. Phacoemulsification and aspiration with intraocular lens insertion (PEA + IOL, phaco group) or MIGS with PEA + IOL (µLot-phaco group) was performed on patients with glaucoma. Age, sex, glaucoma type, pre- and postoperative IOP, and ratio of IOP reduction were estimated. **Results**: The rate of IOP reduction in the µLot-phaco group was found to be significantly higher than in the phaco group at the 6-month postoperative assessment. Specifically, a strong correlation was observed between preoperative IOP levels, the presence of POAG, and patient age within the µLot-phaco group, all of which contributed significantly to the IOP reduction observed at the 6-month follow-up. **Conclusions**: Simultaneous cataract and MIGS were found to be more effective in older POAG patients with higher preoperative IOP.

## 1. Introduction

Microhook ab interno trabeculotomy (µLot) is a well-established minimally invasive glaucoma surgery (MIGS) technique, recognized for its efficacy in reducing intraocular pressure (IOP) [1,2,3]. While cataract surgery alone (phaco) can lower IOP, combining cataract surgery with µLot (µLot-phaco) has emerged as a superior intervention for IOP reduction in glaucoma patients [4,5,6]. Prior studies have identified potential complications associated with µLot, including hypotony and significant hyphema following acute IOP spikes, which may require additional glaucoma filtration surgery [7,8,9,10].

Despite advancements in minimally invasive procedures like MIGS, current glaucoma treatments have limitations. Challenges such as IOP variability, potential complications from surgeries, and the need for ongoing medical therapy present significant obstacles to achieving long-term management success in glaucoma patients. While MIGS has transformed surgical approaches by providing less invasive options, the effectiveness of these procedures in specific patient populations, outcome variability, and the selection of ideal candidates for MIGS continue to pose ongoing challenges in glaucoma care. Understanding these limitations is critical for refining treatment strategies and improving patient outcomes in glaucoma management. Therefore, the careful selection of patients for µLot based on individual characteristics is essential.

This study aims to investigate the effectiveness of µLot-phaco in various glaucoma types and examine postoperative IOP changes compared to phaco alone. By analyzing the outcomes of µLot-phaco in different glaucoma subtypes in detail, this research seeks to offer valuable insights into the optimal management approach for glaucoma patients undergoing cataract surgery. Through evaluating the efficacy and safety of µLot-phaco in diverse glaucoma presentations, we aim to contribute to the existing knowledge base and enhance the clinical decision-making process for treating glaucoma patients with concurrent cataract issues.

## 2. Subjects and Methods

### 2.1. Ethical Approval

This retrospective cohort study was approved by our institutions committee (Tohankai Eye Institution’s Ethics Committee, approval number 0001) and adhered to the regulations of clinical practice and the tenets of the Declaration of Helsinki.

### 2.2. Study Design

Glaucoma patients undergoing cataract surgery were included in this study, which was conducted between June 2018 and June 2019. The cataract surgeries were performed using the Constellation Vision System (Alcon, Tokyo, Japan), located at various eye institutions including Urawa Central Eye Institution, Kawagoe Central Eye Institution, Koshigaya Central Eye Institution, Kumagaya Central Eye Institution, Yorii Central Eye Institution, Onohara Eye Institution, and Kisarazu Central Eye Institution. The surgical procedures were carried out by two experienced surgeons, namely Hayakawa Y and Inada T.

The sample size is decided based on factors like the study goals, the effect size (how big the effect is expected to be), and the desired statistical power (likelihood of not missing a true effect), as well as the values of α (level of significance) and β (probability of making a Type I error).

The patients were divided into two groups: the phaco group, which underwent cataract surgery only, and the µLot-phaco group, which underwent cataract surgery in combination with ab interno trabeculotomy. Throughout the study period, the type and number of ophthalmic antiglaucoma medications remained consistent for each patient before and after the surgery.

Intraocular pressure (IOP) measurements were taken using a non-contact tonometer (manufactured by Tomey, Tokyo, Japan). These measurements were conducted before the surgeries, postoperatively, and during follow-up visits to monitor the changes in IOP levels in both groups of patients. The inclusion of multiple eye institutions and standardized measurement techniques enhances the robustness and generalizability of the study findings regarding the outcomes of cataract surgery alone versus cataract surgery combined with ab interno trabeculotomy in glaucoma patients.

### 2.3. Inclusion and Exclusion Criteria

The study recruited individuals with mild-to-moderate glaucoma, including those diagnosed with primary open-angle glaucoma (POAG), normal tension glaucoma (NTG), and combined mechanism glaucoma (CMG), who also had visually significant cataracts and were using one or more ophthalmic antiglaucoma medications [11]. Diagnosis of POAG was based on the presence of characteristic glaucomatous optic neuropathy, two reliable visual field tests showing repeatable glaucomatous defects, an open angle observed on gonioscopy, and an intraocular pressure (IOP) consistently higher than 21 mmHg on two consecutive visits using a Goldman applanation tonometer. In contrast, NTG was diagnosed based on similar criteria but with an IOP consistently below 21 mmHg. Patients not meeting the criteria for a specific type of glaucoma such as POAG or chronic angle-closure glaucoma (CACG) were categorized as having CMG. Eligible participants for the study were individuals with cataracts and glaucoma, aged 18 years or older, who had an IOP below 30 mmHg. 

Exclusion criteria encompassed specific types of glaucoma, such as exfoliation glaucoma, secondary glaucoma, and neovascular glaucoma. Furthermore, individuals with cataracts and high myopia characterized by an axial length exceeding 27 mm were excluded from this study to ensure homogeneity within the study population. Additionally, patients with complications during cataract surgery, such as posterior lens capsule rupture and zonular dehiscence, were also excluded to prevent confounding factors that could impact the outcomes of the study. By carefully selecting participants based on detailed inclusion and exclusion criteria, the study aimed to maintain a consistent and well-defined study population to accurately evaluate the effects of cataract surgery, both with and without ab interno trabeculotomy, on individuals with mild-to-moderate glaucoma and visually significant cataracts.

### 2.4. Surgical Procedure

Before the commencement of the surgical procedure, patients received topical eye drops and intracameral anesthesia containing 4% and 1% lidocaine, respectively. The ab interno trabeculotomy, known as µLot, was executed using a straight-type Tanito microhook specifically designed for ab interno trabeculotomy (M-2215, Inami, Tokyo, Japan). This microhook was meticulously utilized to excise the trabecular meshwork at both the nasal and temporal quadrants, precisely performed following the creation of a continuous curvilinear capsulorhexis (CCC) utilizing a Swan Jacob Gonioprism lens (R E MEDICAL, Osaka, Japan).

The subsequent steps of the procedure involved phacoemulsification and aspiration with intraocular lens implantation (PEA + IOL), all carried out through a 2.4 mm corneal incision made temporally using the high-precision Constellation Vision System (Alcon). This comprehensive surgical approach aimed to not only address the cataract but also incorporate ab interno trabeculotomy to enhance surgical outcomes and potentially reduce intraocular pressure postoperatively in glaucoma patients. The meticulous execution of each step using advanced instrumentation and techniques underscores the precision and thoroughness of the surgical intervention conducted by the experienced surgical team.

In this study, the calculation of IOL power for cataract surgery was performed using the Barrett Universal II formula in all cases. The Barrett Universal II formula is a thick lens formula based on Gaussian optics, and although the details are not publicly disclosed, it is known to use an optical model consisting of two spheres: one reflecting the anterior segment incorporating corneal back surface shape, and the other reflecting the posterior segment incorporating the eye’s shape. It has gained popularity as a formula, surpassing the SRK/T formula, and has been widely adopted in cataract surgeries [12].

### 2.5. Measurement

Various clinical parameters were meticulously documented and retrieved from the medical records of the patients undergoing cataract surgery in conjunction with glaucoma management. These parameters included demographic details such as age, gender, the specific type of glaucoma, the presence or absence of a history of diabetes mellitus, axial length (AL), equivalent spherical power, preoperative and postoperative intraocular pressure (IOP) readings, as well as whether argon laser peripheral iridoplasty (ALPI) was performed. The ratio of IOP reduction, calculated as (preoperative IOP − postoperative IOP/preoperative IOP) × 100, was also a critical parameter assessed in this study.

Axial length measurements were obtained using the IOL Master 500 (Carl Zeiss Meditec, Dublin, CA, USA), while the evaluation of equivalent spherical power was conducted using a ref-keratometer (Tomey). In cases where there was gonio adhesion, ALPI was carried out one week post ab interno trabeculotomy (µLot) to prevent the formation of anterior synechiae. The estimation of IOP was performed utilizing a non-contact tonometer (Tomey).

A total of 159 eyes that underwent cataract surgery in combination with glaucoma treatment were included in the analysis. These eyes were longitudinally monitored for a duration exceeding 6 months following the surgical intervention, during which the number of instillations for managing glaucoma remained consistent. Among these cases, the surgical approach comprised cataract surgery combined with minimally invasive glaucoma surgery in 95 eyes (referred to as the µLot-phaco group), whereas cataract surgery alone was conducted in the remaining 64 eyes (referred to as the phaco group). Table 1 and Table 2 present a detailed overview of the clinical characteristics and background of both distinct groups, providing valuable insights into the patient profiles and outcomes associated with the different surgical strategies employed.

### 2.6. Statistical Analysis

All statistical analyses were performed using EZR software (version 1.55) [13]. The normality of the univariate analysis in less than 30 eyes was assessed using a Kolmogorov–Smirnov test. The normality assumption for more than 30 eyes was confirmed using histogram distribution. The homoscedasticity was evaluated using an F test between two groups and a Bartlett test between three groups, respectively. The comparison for the rate of IOP reduction between the phaco and the µLot-phaco groups was analyzed with a *t*-test (Welch) after the F test. A multiple regression analysis was carried out to evaluate the independent influencing factor after organizing the confounders. The validity of multiple regression analysis was confirmed by a quantile-quantile plot (Q-Q plot), which indicates residual normality.

## 3. Results

### 3.1. Demographic

One hundred fifty-nine eyes with cataract and glaucoma who underwent phaco or µLot-phaco were included in the study analysis. Demographics and preoperative parameters are shown in Table 1 and Table 2.

### 3.2. The Comparison of IOP Reduction Rate 

Despite observing a similar impact between the phaco group and the µLot-phaco group on the rate of intraocular pressure (IOP) reduction at 1 and 3 months post-surgery, it was noteworthy that the rate of IOP reduction in the µLot-phaco group exhibited a significantly higher magnitude compared to the phaco group at the 6-month postoperative mark (*p* = 0.03005) (Figure 1).

### 3.3. Factor of IOP Reduction in µLot-Phaco Group

The statistical analysis methods, specifically univariate analysis, employed in the µLot-phaco group are outlined in Table 3. A Q-Q plot was utilized to demonstrate the suitability of multiple regression analysis in the µLot-phaco group, as indicated in Figure 2. Among the four continuous variables considered, preoperative IOP exhibited a significant correlation with the rate of IOP reduction at the 6-month postoperative time point (r = 0.372, *p* = 0.000203), illustrated in Figure 3. Notably, no significant differences were observed in the analysis of the three categorical variables, namely sex, diabetes mellitus, and postoperative ALPI. 

Moreover, a comprehensive multiple linear regression analysis was conducted to explore the relationship between IOP reduction and various clinical parameters mentioned earlier (as described in the Subjects and Methods section). The results revealed that preoperative IOP, primary open-angle glaucoma (POAG), and age were significantly associated with IOP reduction at the 6-month postoperative assessment (*p* = 0.00000745, *p* = 0.000945, *p* = 0.001138, respectively, Table 4). This suggests that there is a compelling case for considering older patients with POAG and elevated IOP as suitable candidates for ab interno trabeculotomy combined with PEA + IOL. 

Based on the above results, we present a flowchart in Figure 4 outlining the treatment approach for cataract surgery in cataract patients with concomitant glaucoma.

## 4. Discussion

The ab interno trabeculotomy technique, including microhook ab interno trabeculotomy (µLot), has captured the interest of glaucoma surgeons due to its technical ease and comparable effectiveness in lowering IOP compared to the traditional ab externo approach [14,15,16,17]. While the ab externo method is effective, its limitations, such as the need for conjunctival incisions that may complicate future filtration glaucoma surgeries, have paved the way for the adoption of the promising µLot technique in managing IOP in glaucoma patients.

Moreover, studies have shown that cataract extraction can reduce IOP in various types of glaucoma, especially in cases of mixed glaucoma [18,19]. However, the combination of cataract surgery with procedures like µLot poses challenges, such as potential complications including decreased visual acuity due to hypotony, hyphema, and IOP fluctuations [20,21]. This raises the critical question of the optimal treatment approach for glaucoma patients with coexisting cataracts—whether standalone cataract surgery or a combined approach with µLot is more beneficial, warranting further investigation.

In our study involving a randomized cohort of 159 eyes, we identified factors influencing IOP reduction following cataract surgery alone or with µLot. Despite potential discrepancies in the study design that could limit result interpretation, our findings highlighted the consistent use of glaucoma medications pre- and post-surgery even after IOP reduction, providing valuable insights into postoperative IOP dynamics. While our study showed a slightly lower rate of IOP reduction with µLot than previous reports [1,2], this difference may be attributed to the delicate microhook approach employed in our technique. Multiple linear regression analysis indicated the relevance of IOP reduction rate, preoperative IOP, POAG, and age, suggesting the efficacy of µLot in elderly POAG patients with elevated IOP. Considering the less favorable outcomes of trabeculectomy in older POAG patients in previous studies, our findings underscore the importance of µLot in managing elderly POAG individuals [22,23].

Looking ahead, future research could focus on enhancing the µLot technique by exploring expanded angles of approach and developing modified instrument designs for improved outcomes. Investigating the long-term effects of combining cataract surgery with µLot on visual outcomes and glaucoma progression could provide significant insights for clinical practice. Additionally, there is potential to explore the application of µLot in other forms of glaucoma, such as developmental and secondary glaucomas, to broaden its utility. Understanding the applicability of µLot in specific patient populations, including those with advanced glaucoma or complex cases, may pave the way for personalized treatment strategies in glaucoma management.

In conclusion, as we advance towards personalized approaches in managing glaucoma, continued exploration of the µLot technique remains vital. Through rigorous studies and methodological refinements, we aim to enhance outcomes for glaucoma patients and elevate the standard of care in ophthalmology. By fostering collaborative efforts and a dedication to research, we strive to improve the adoption of innovative techniques like µLot for the benefit of glaucoma patients globally.

## Figures and Tables

**Figure 1 jcm-13-05378-f001:**
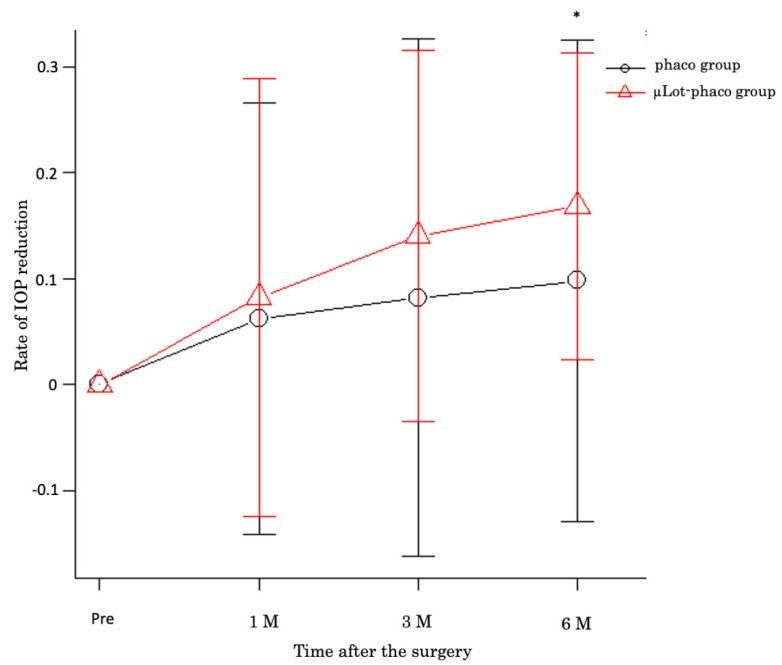
The rate of IOP reduction in µLot-phaco group at 1, 3, and 6 months after the surgery. Significant IOP reduction rate observed in 6 M after the surgery * *p* = 0.03005.

**Figure 2 jcm-13-05378-f002:**
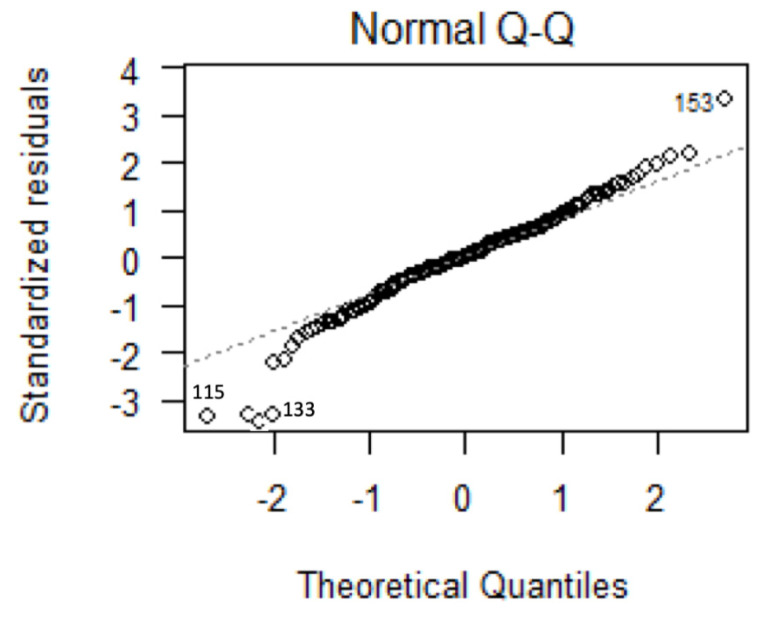
Multiple linear regression analysis (159 eyes): residuals Q-Q plotting.

**Figure 3 jcm-13-05378-f003:**
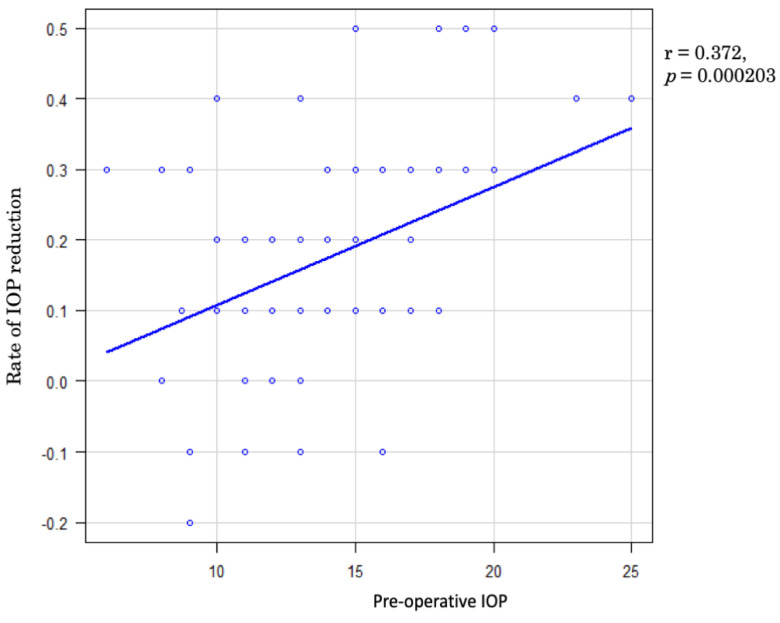
Rate of IOP reduction at 6 months after the surgery and preoperative IOP were analyzed by Pearson’s correlation coefficient. Rate of IOP reduction was correlated with preoperative IOP (r = 0.372, *p* = 0.000203).

**Figure 4 jcm-13-05378-f004:**
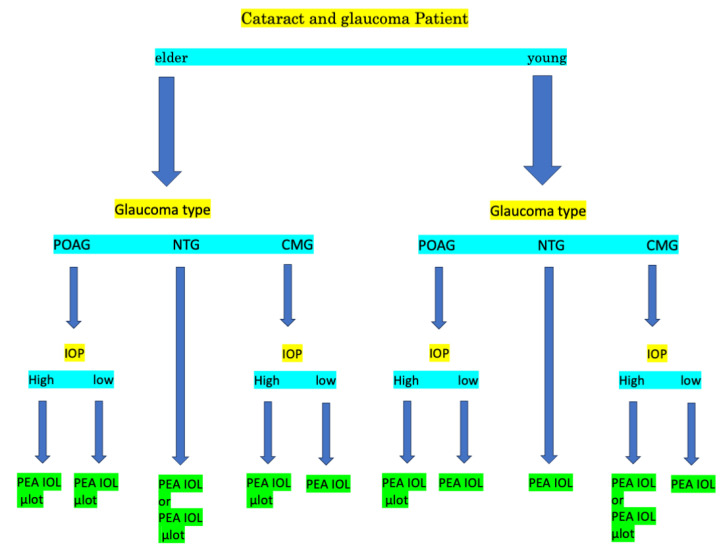
Flow chart of the treatment of cataract surgery with or without μLot of patients with cataract with glaucoma.

**Table 1 jcm-13-05378-t001:** Phaco group.

Category	Group	*n* (Total 64 Eyes)
sex	male	35
	female	29
age	40.3~93.8 (74.3 ± 8.8) years old	
Diabetes Mellitus	with	7
	without	57
axial length	20.9~28.6 (23.8 ± 1.4) mm	
spherical equivalent	−14.0~+4.9 (−1.2 ± 3.9) D	
pre-operative IOP	5.0~27.0 (13.0 ± 3.5) mmHg	
glaucoma type	POAG	31
	NTG	18
	CMG	15

**Table 2 jcm-13-05378-t002:** µLot-phaco group.

Category	Group	*n* (Total 95 Eyes)
Sex	male	47
	female	48
age	44.9~92.1 (74.6 ± 8.2) years old	
Diabetes Mellitus	with	17
	without	78
axial length	21.2~27.9 (23.9 ± 1.4) mm	
spherical equivalent	−13.8~+8.4 (−1.4 ± 3.8) D	
pre-operative IOP	6.0~25.0 (13.6 ± 3.2) mmHg	
glaucoma type	POAG	56
	NTG	21
	CMG	18
post-operative LGP	with	24
	without	71

**Table 3 jcm-13-05378-t003:** Statistical analysis methods (univariate analysis) adopted in µLot-phaco group.

Category	*n*	Normality	Equal Variance	Statistical Analysis
age	95	◯ assumption		Peason’s correlation coefficient
axial length	95	◯ assumption		Peason’s correlation coefficient
spherical equivalent	95	◯ assumption		Peason’s correlation coefficient
pre-operative IOP	95	◯ assumption		Peason’s correlation coefficient
Sex	M 47	◯ assumption	◯	t-test
	F 48	◯ assumption		
Diabetes Mellitus	With 17	◯ test	◯	t-test
	Without 78	◯ assumption		
glaucoma type	POAG 56	◯ test	✕	ANOVA (welch)
	NTG 21	◯ test		
	CMG 18	◯ test		
post-operative LGP	with 24	◯ test	◯	t-test
	without 71	◯ assumption		
operative method	Cat μlot 95	◯ assumption	✕	t-test (welch)
	Cat 64	◯ assumption		

Equal variance indicates the property of whether the variation (variance) of data is the same between different groups in statistics. Using symbols, ◯ represents “Equal variance, not rejected between the two groups”, while ✕ represents “Variances rejected between the two groups”.

**Table 4 jcm-13-05378-t004:** Multiple linear regression analysis of µLot-phaco group.

Category	Estimate	Std. Error	t-Value	Pr(>|t|)
(Intercept)	−0.28035	0.498885	−0.562	0.575633
gla.type [NTG]	0.071901	0.044845	1.603	0.112579
gla.type [POAG]	0.133543	0.038977	3.426	0.000945 ***
pre-ope IOP	0.021706	0.004547	4.774	7.45 × 10^−6^ ***
axial length	−0.01794	0.021215	−0.846	0.400049
post-ope LGP [+]	−0.0097	0.031723	−0.306	0.76053
age	0.006263	0.001859	3.368	0.001138 ***
sex	0.026498	0.030577	0.867	0.388597
Spherical equivalent	−0.00607	0.007156	−0.848	0.398794
Diabetes mellitus [+]	0.004731	0.037554	0.126	0.900042

In multiple linear regression, when the *p*-value associated with a specific predictor variable is less than 0.05 (***), it is considered statistically significant, suggesting that there is a strong evidence that the predictor variable affects the response variable.

## Data Availability

The datasets generated and analyzed during the current study are available from the corresponding author on reasonable request.

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
