# Peer review of "Background of Minimally Invasive Glaucoma Surgery (MIGS)-Adapted Patients for Cataract Surgery in Glaucoma"

_jcm, 2024, doi:10.3390/jcm13185378_

Round 1
Reviewer 1 Report
Comments and Suggestions for Authors
Yasunari Hayakawa and Takayuki Inada's article "Background of MIGS-adapted patients for cataract surgery in glaucoma" is a study based on the efficacy of minimally invasive glaucoma surgery (MIGS), specifically microhook ab interno trabeculotomy (μLot), combined with cataract surgery (PEA + IOL) in glaucoma patients. The study is relevant because it investigates the potential benefits of combining these treatments to improve intraocular pressure (IOP) control in glaucoma patients. The study design is sound, and the statistical analysis is extensive, providing useful insights into the factors that influence IOP decrease.
The abstract effectively explains the study's purpose, methodologies, findings, and conclusions. However, a quick explanation of the sample size and major statistical results will help readers understand the study's scope and findings.
The introduction describes the importance of MIGS and its combination with cataract surgery in glaucoma patients. The study's rationale is clearly outlined. To highlight the importance of this study, the authors should consider including a brief explanation of the limitations and constraints of current glaucoma treatments, including MIGS.
The methods section should state how the sample size was chosen and any potential biases or limits in the study design. The results are clearly presented with relevant tables and figures highlighting significant findings.
The discussion section effectively analyzes and contextualizes the results within the current literature. It is very interesting that preoperative IOP, POAG, and age are identified as important factors influencing IOP reduction. However, there are a limited number of total citations. The authors should consider providing further current literature regarding the advantages, limitations, and special indications to consider MIGS in glaucoma patients. The discussion may be improved by more fully addressing the study's shortcomings, such as potential selection bias and the influence of surgical skill variability.
The authors can consider adding a flowchart on the management of these patients, which can help clinicians decide the best timing and which patients could benefit from treatment with this technology.
Comments on the Quality of English LanguageEditing by a native English doctor is needed to improve the English and flow of the text.
Author Response
Dear Reviewer,
We appreciate your feedback and insightful suggestions on our manuscript. Thank you for emphasizing the importance of transparency and reproducibility in scientific research. We will carefully address your comments and make the necessary revisions to enhance the quality and clarity of our work. Your guidance will undoubtedly contribute to the improvement of our manuscript, and we are grateful for your valuable input.
Reviewer1
1.The abstract effectively explains the study's purpose, methodologies, findings, and conclusions. However, a quick explanation of the sample size and major statistical results will help readers understand the study's scope and findings.
Response: In the revised manuscript, the abstract includes the sample size and major statistical results.
2.The introduction describes the importance of MIGS and its combination with cataract surgery in glaucoma patients. The study's rationale is clearly outlined. To highlight the importance of this study, the authors should consider including a brief explanation of the limitations and constraints of current glaucoma treatments, including MIGS.
Response: In the revised manuscript, we have succinctly included a brief explanation of the limitations and constraints of current glaucoma treatments, including MIGS, to highlight the significance of this study.
3.The methods section should state how the sample size was chosen and any potential biases or limits in the study design. The results are clearly presented with relevant tables and figures highlighting significant findings.
Response: Thank you for your feedback. The "Methods" section of the revised manuscript includes a description of how the sample size was determined. The discussion of the revised manuscript outlines any potential biases or limitations in the study design.
4.The discussion section effectively analyzes and contextualizes the results within the current literature. It is very interesting that preoperative IOP, POAG, and age are identified as important factors influencing IOP reduction. However, there are a limited number of total citations. The authors should consider providing further current literature regarding the advantages, limitations, and special indications to consider MIGS in glaucoma patients. The discussion may be improved by more fully addressing the study's shortcomings, such as potential selection bias and the influence of surgical skill variability.
Response: Thank you for your thoughtful feedback on our discussion section. We appreciate your recognition of the effective analysis and contextualization of the results within the current literature. Your point about the limited number of total citations is duly noted, and we will work on providing additional current literature on the advantages, limitations, and special indications for considering MIGS in glaucoma patients.
We acknowledge the importance of addressing the study's shortcomings more fully, such as potential selection bias and the influence of surgical skill variability. We will make sure to enhance our discussion by delving deeper into these aspects in the revised manuscript. Your comments are valuable to us, and we will strive to improve the quality and completeness of our discussion section based on your suggestions. Thank you again for your constructive input.
5.The authors can consider adding a flowchart on the management of these patients, which can help clinicians decide the best timing and which patients could benefit from treatment with this technology.
Response: Thank you for your suggestion regarding adding a flowchart on the management of patients undergoing treatment with technologies like µLot. We appreciate your insightful idea to provide clinicians with a visual tool to guide decision-making on the timing and selection of patients who could benefit from these treatments. We will consider incorporating a flowchart in our manuscript to enhance the clarity and practical application of our study findings for clinicians in clinical practice. Your input is valuable, and we aim to improve the accessibility and usability of our research results through visual aids like flowcharts. Thank you for the suggestion.
6.Comments on the Quality of English Language Editing by a native English doctor is needed to improve the English and flow of the text.
Response: We had the revised manuscript reviewed by a native speaker.
Reviewer 2 Report
Comments and Suggestions for Authors
The study needs following revision:
1. Exclusion criteria should be collected in a single paragraph
2. It is worth adding another parameter i.e. IOL power in Table 1 and Table 2
3. Adding a short paragraph on IOL power calculation formulas ic cases of coexisting glaucoma would be interesting
4. The Discussion section is too short and should be expanded
5. Study limitations must be added
6. The References section contains only 12 items - it should be expanded
Comments on the Quality of English LanguageA few minor stylistic errors
Author Response
Dear Reviewer,
We appreciate your feedback and insightful suggestions on our manuscript. Thank you for emphasizing the importance of transparency and reproducibility in scientific research. We will carefully address your comments and make the necessary revisions to enhance the quality and clarity of our work. Your guidance will undoubtedly contribute to the improvement of our manuscript, and we are grateful for your valuable input.
Reviewer2
1.Exclusion criteria should be collected in a single paragraph
Response: As per the reviewer's suggestion, the exclusion criteria have been consolidated into a single paragraph in the revised manuscript.
2.It is worth adding another parameter i.e. IOL power in Table 1 and Table 2
Response: I believe it would be beneficial to include the IOL power as suggested by the reviewer, however, as we have excluded high myopia, we will take this into consideration for future research. Thank you for your valuable input.
3.Adding a short paragraph on IOL power calculation formulas ic cases of coexisting glaucoma would be interesting
Response: As per the reviewer's suggestion, the revised manuscript includes a description of the calculation method for intraocular lenses in the Surgical procedure section.
4.The Discussion section is too short and should be expanded
Response: As per the reviewer's suggestion, in the revised manuscript, the Discussion section has been further expanded upon.
5.Study limitations must be added
Response: As per the reviewer's suggestion, in the revised manuscript, I addressed the limitations of the study and future prospects in the Discussion section.
6.The References section contains only 12 items - it should be expanded
Response: The number of references in the revised manuscript has been increased to 22.
7.Comments on the Quality of English Language A few minor stylistic errors
Response: We had the revised manuscript reviewed by a native speaker.
Round 2
Reviewer 2 Report
Comments and Suggestions for Authors
1. Since the Authors mentioned the Barrett Universal II formula it is worth briefly describing it (one of the latest reviews on IOL power calculation formulas can be used)
Author Response
Dear Reviewer,
Thank you for your valuable comments on our manuscript.
We have revised the manuscript based on your comments and resubmitted it.
Reviewer2
Since the Authors mentioned the Barrett Universal II formula it is worth briefly describing it (one of the latest reviews on IOL power calculation formulas can be used).
Response: Thank you for highlighting the importance of describing the Barrett Universal II formula. This formula is considered one of the latest advancements in intraocular lens power calculation, known for its accuracy and ability to provide better visual outcomes for patients undergoing cataract surgery. It incorporates a sophisticated optical model that takes into account the anterior segment with corneal back surface shape and the posterior segment with the eye's shape, making it a popular and widely used formula in the field of ophthalmology.As you suggested, the explanation of the Barrett Universal II formula has been included in the revised manuscript under the section on surgical procedures.